# Better coverage, better outcomes? Mapping mobile network data to official statistics using satellite imagery and radio propagation modelling

**Till Koebe** * 

Department of Economics, Freie Universität, Berlin, Germany

* till.koebe@fu-berlin.de

## Abstract

Mobile sensing data has become a popular data source for geo-spatial analysis, however, mapping it accurately to other sources of information such as statistical data remains a challenge. Popular mapping approaches such as point allocation or voronoi tessellation provide only crude approximations of the mobile network coverage as they do not consider holes, overlaps and within-cell heterogeneity. More elaborate mapping schemes often require additional proprietary data operators are highly reluctant to share. In this paper, I use human settlement information extracted from publicly available satellite imagery in combination with stochastic radio propagation modelling techniques to account for that. I show in a simulation study and a real-world application on unemployment estimates in Senegal that better coverage approximations do not necessarily lead to better outcome predictions.

**Data Availability Statement:** The mobile phone data at the antenna- and commune-level aggregated to the year 2013 including noisy

## Introduction

Mobile phone metadata has become a popular data source to complement official statistics. When an individual makes a call, sends a message or uses the mobile internet, meta information about this interaction, such as the time stamp and the location, are stored in a database of the mobile network operator (MNO). Researchers exploit those spatio-temporal references for geo-located analysis. One string of research in this field investigates the question whether a certain characteristic such as poverty, literacy or food insecurity is reflected in mobile phone behaviour. Matching this behaviour accurately to a 'groundtruth'—often statistical data from surveys or censuses provided for statistical areas—however, poses a major challenge as the two data sources lack a common reference. In the case of call detail records (CDRs), the geographic reference is provided by the antenna location, often stored as a point coordinate of the physical location of the corresponding base transmitter station (BTS). Due to its simplicity, some scientific literature treat antennas as point coordinates [1]. However, the interactions captured by the antenna do not happen entirely at this exact coordinate, but within the coverage area of the antenna—the cell. While an antenna may be located in one statistical area, most of the cell may lie within the neighboring area. The state-of-the-art attempt to address this is to use spatial weights based on the overlapping area size of statistical areas and cells approximated via voronoi tessellation [2, 3]. This approach has three major drawbacks: First, voronoi tessellation

antenna locations as well as instructions for replicating the study results have been added as part of the Supporting information. In order to access record-level mobile phone data and exact antenna locations, one would need to contact Sonatel directly and present the research project that would require the data (contact: Mr El Hadji Birahim Gueye, Direction des Systèmes d'information Sonatel, ebgueye@orange-sonatel. com or post mail: Orange-Sonatel, 46 Boulevard de la République, BP 69 Dakar, Senegal). GUF data cannot be shared publicly because third-party access conditions apply (for scientific, non-commercial use). However, it is available for research purposes under a data user agreement. For data access, please contact the German Aerospace Agency under guf@dlr.de (https://www. dlr.de/eoc/en/PortalData/60/Resources/ dokumente/guf/DLR-GUF_LicenseAgreement-and-OrderForm.pdf). Census data used in the study cannot be shared publicly because third-party access conditions apply. However, it is available for research purposes under a data user agreement. For data access, please visit the microdata catalogue of the statistical office in Senegal (http:// anads.ansd.sn/index.php/catalog/51) or send the inquiry to statsenegal@ansd.sn. All code required for replicating the findings of this study is fully available in the Supporting information of this submission (S1 and S2 Files) and under https:// github.com/tilluz/geomatching_open.

**Funding:** The author received no specific funding for this work.

**Competing interests:** The author have declared that no competing interests exist.

perfectly divides the space around BTS locations depending on the distance to the surrounding BTS. This represents a naïve approximation of the true coverage areas as it does not take overlaps, areas without coverage and additional network complexities (multiple antennas per site/ BTS, directionality of antennas, varying frequency bands etc.) into account [4]. For example, roughly 90 million people in Africa in 2019 were still not connected to any mobile network hinting at major holes in the coverage [5]. Second, even though the concept of 'home-locating' subscribers to specific BTS offers a network-based alternative to the statistical concept of 'usual place of residence', it is not reflected within cells. As the weights are based on area sizes, the voronoi tessellation implicitly assumes that individuals/households are homogeneously distributed within cells, which in most cases does not hold true. For example, a lake would receive the same importance in the creation of area-level mobile phone metadata aggregates as an equally sized built-up area. Third, as mobile stations (MS, generally defined as a combination of device and SIM card) and antennas communicate via modulated radio signals whose propagation paths depend on a range of factors such as the weather, coverage areas are stochastic by nature. More elaborate approaches to model coverage ranges of mobile networks exist [4, 6], especially in the field of radio propagation modelling native to electrical engineering, however, they often require detailed information on the area's topology, a number of technical details concerning the network infrastructure and additional information from passive monitoring systems, which mobile network operators are generally highly reluctant to share and in the latter case often not capable to collect.

## Contributions

Acknowledging this, I divide my methodological contribution in this paper in two parts: First, I propose the use of settlement information extracted from publicly available satellite imagery to account for within-cell heterogeneity within the mobile network when linking statistical data with mobile phone metadata. Building on this, the second part of the methodology takes advantage of scenarios where additional technical specifications are available in order to address the issues for holes, overlaps and non-linearities within the mobile network using propagation-based modelling. My main contributions are as follows:

1. The idea of using settlements retrieved from publicly available satellite imagery as a common reference for statistical units such as households and 'home-located' MS in order to calculate weights for mapping mobile phone metadata and statistical data based on settlement counts in scenarios where MS counts are not available. This way, within-cell heterogeneity is addressed.

2. A propagation-based approach to account for overlaps, holes and non-linearities in coverage service provision—in case additional information on the network infrastructure are available.

3. A large-scale simulation study on a synthetic population grid to systematically compare the accuracy of different mapping approaches and their effects on predictive performance.

4. A real-world application that demonstrates the impact of the mapping choice on outcomes in later analysis.

## Datasets

In the application, I revisit the simulation study of Schmid et al. [1] published in 2017 in the *Journal of the Royal Statistical Society Series A* on fine-granular unemployment estimates from mobile phone metadata in Senegal in order to investigate the effects of different mapping schemes on the unemployment outcomes. Therefore, I re-run the original simulation with the difference that I implement multiple mapping schemes to derive area-level covariates from

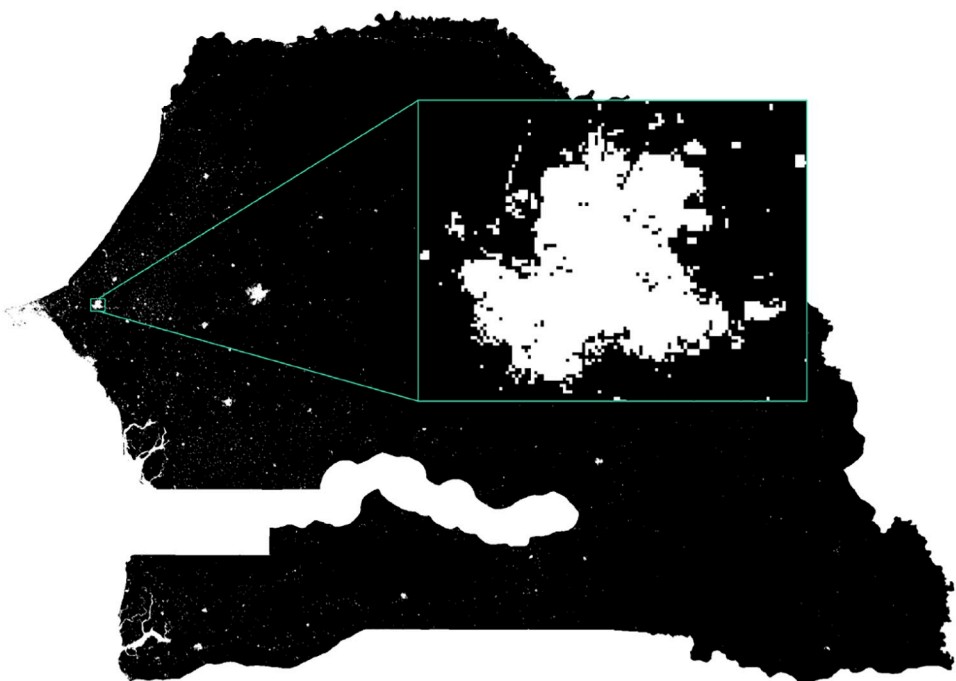

**Fig 1. Settlements in Senegal provided as b/w image by the GUF project.** Lower resolution built-settlements extents data reprinted from [10] under a CC BY license, with permission from WorldPop, original copyright 2018, are used in this figure for illustrative purposes.

CDRs. Specifically, I use behavioural indicators and SIM card counts extracted from CDRs provided by the major Senegalese MNO *Sonatel* in the context of the D4D 2014 challenge for the whole year of 2013 and aggregated on the level of BTS, for which the exact geo-coordinates are also provided [7]. The behavioural indicators are generated using the popular open-source Python module *Bandicoot* [8]. Further, I use population counts from the full 2013 general population and housing census (*RGPHAE 2013*) available for the NUTS 4-level of Senegal—the *communes*—on the website of *ANSD*, the National Statistical Office of Senegal. Commune-level unemployment information are generated from a 10% sample of RGPHAE 2013. Unemployment information in RGPHAE 2013 are self-reported.

Geographic information on the administrative boundaries are available for communes and above. The settlement-based weights I present in this paper use data on human settlement areas in Senegal extracted from the Global Urban Footprint (GUF) project [9] of the German Aerospace Center (DLR) at a resolution of 0.4 arc seconds, which is approximately 12m x 12m. The GUF project used 180,000 TerraSAR-X and TanDEM-X images collected during the period of 2011—2012 (with some data from 2013/14 to fill gaps) to create black and white abstractions where white pixels represent human settlements with a true positive rate (accuracy to correctly detect human settlements) of 85% on average, with 68% at lowest and 98% at heighest. GUF data for Senegal is provided as a single black and white.tif-file with a resolution of 55568 x 39459 pixels (see Fig 1). All datasets used in this study are available for research purposes under the conditions of the respective data use agreements.

## Related work

Increasing processing capabilities have propelled the use of satellite imagery in official statistics. The UN [11] recommends using satellite imagery to prioritize and check geospatial

processes such as the delineation of enumeration areas during census preparation. It further supports the construction of population grids as a common spatial reference system as proposed by [12, 13]. Various studies have used remote sensing, sometimes in combination with mobile phone metadata, to estimate key statistical indicators such as economic growth [14–16], population density [17–20] or poverty [2, 21, 22]. Work in that field most closely related to this study uses settlement information extracted from satellite imagery in combination with radio propagation models for application in cost-benefit analysis concerning additional infrastructure investments [23]. While [23] also uses population counts from official statistics to estimate the latent demand for mobile services, the author neither investigates the effects of different coverage mapping techniques on the results nor does he use mobile phone metadata for statistical purposes.

In addition, the last decade has seen an impressive amount of research on proposing the use of mobile phone metadata for official statistics foremost in the hope to overcome the limiting relationship of sample size and data collection costs. [24] provides an excellent overview on the use of mobile phone metadata that also covers its application for statistical purposes. Use cases to produce more frequent, more granular and/or more timely data on a wide range of statistical topics have been identified. For example, [4, 25–28] use mobile phone metadata to investigate population dynamics for more frequent population and tourism statistics. [29, 30] apply the question on the whereabouts of a population to the post disaster setting. Mobility aspects such as commuting and travelling routines have been looked at in more detail by [31–36]. By exploiting both mobility and (social) network characteristics of mobile phone metadata, [37–42] and [43, 44] use mobile phone metadata to model disease spreading and integration, respectively. Mobile usage patterns have been explored to provide fine granular insights on socio-demographic indicators such as multi-dimensional poverty [2, 3], literacy [1, 45] and economic vulnerability [46, 47]. While most of these studies have mapped mobile phone metadata and groundtruth data using point-to-polygon allocation or voronoi tessellation, very few studies have applied more elaborate approximation schemes. [4] propose a methodology based on maximum likelihood estimation that uses cell footprints provided by one or multiple MNOs in combination with location data from passive monitoring systems to acquire more accurate measures on the density of MS. The authors run a simulation study on a 100x100m synthetic population grid to compare the proposed methodology against voronoi-based coverage maps. However, the methodology requires very detailed information from the involved MNOs, e.g. on the cell footprints and the signalling data that may prove difficult to acquire in practice (see Section Mobile phone metadata). Further, while the authors rightly assume a multinomial distribution of the MS counts, finding appropriate distributions for the wide range of behavioural covariates appears less trivial. In order to simplify and improve the coverage mapping process, members of the European Statistical System as part of the *ESSnet Big Data* project are currently developing *mobloc* [48]—an *R* package that implements the free space path loss propagation model using technical specifications of antennas as input parameters. However, neither [4] nor [48] systematically evaluate different coverage mapping techniques on statistical modelling approaches using real-world data.

## Background

### Mobile phone metadata

Mobile networks not only transport data for communication purposes, they also generate data for reasons such as network auditing, billing, maintenance and service provision. Some of this meta information is created in interaction with user equipment such as MS. There are four main caveats of using mobile phone metadata for population statistics in general. All of them

have in common that they are active areas of current research. First, the customer base of an MNO constitutes a non-representative population sample with unknown sampling design. The consequences are varying sampling rates, i.e. locally changing market shares and parts of the population being structurally excluded from the sample such as children, elderly and the very poor. Second, the unit of observation—i.e. the MS, device, the SIM card and/or the subscriber— does not perfectly match the unit of interest, which is the individual or household, as phone sharing schemes or multi-SIM uses illustrate. Common approaches to account for these two caveats are calibration and/or reconstructing the sampling design empirically. Third, mobile phone metadata lacks the statistical concept of *usual residence*—a concept frequently used in official statistics to determine the geo-location of an individual/household defined as the place where an individual has lived or intends to live for a period of at least 6 or 12 months [49]. Different approaches to approximate the *home location* of an MS exists (e.g. night-time home location defined as the most frequently used cell by an MS between 7pm and 7am during a certain time window), however, the definitions do not map perfectly introducing uncertainty in further analysis [50]. Fourth, coverage areas cannot be pinpointed as radio propagation is dynamic and stochastic by nature. Propagation models of various complexity exist to provide approximations as coverage ranges can generally vary from couple of hundred meters to over 40km.

Most scientific studies in the context of international development and official statistics use CDRs—logs of interactions such as calls, text messages or internet use containing attributes of the MS, the network and the connection—as a basis for further analysis. The advantages of CDRs compared to other mobile phone metadata such as Visitor Location Registers (VLRs) or other signalling data are threefold: First, they provide fine-grained geographical resolution through cell-level identifiers. Second, they provide information both on the mobility and the (social) network of the MS. Third, CDRs are fairly easy to access and to use in analysis as the storage of essential attributes adheres to global standards such as *3gpp 32.295*. However, in addition to the aforementioned general caveats of mobile phone metadata there are important caveats specific to CDRs: Social network information extracted from CDRs are increasingly incomplete due to a shift towards app-based communication (e.g. Whatsapp and Facebook messenger). Mobility patterns are fragmented as locations are logged only during active MS use— again a case of non-random sampling. Some MNOs are able to extract more detailed information on the location of an MS and its app usage e.g. for geo-fencing purposes or app-based pricing schemes through trilateration of signalling data and deep packet inspection, respectively. This, however, requires specific hardware equipment and software capabilities, which not every MNO has. Consequently, these type of information are rarely available to researchers.

## Radio propagation modelling

Radio propagation modelling has been subject to research for decades. Coverage mappings in mobile networks are generally used for network planning purposes [23, 51]. Looking at Phillips et al. [6] is highly recommended as they provide an excellent overview on coverage mapping methods. In general, radio propagation modelling techniques in mobile networks largely focus on estimating the path loss $L_p$ a radio signal incurs en route between a transmitter *tx* and a receiver *rx*. Together with the output power of the transmitter $P_{tx}$, the gains through directivity and efficiency of the involved antennas $G_{tx}$ and $G_{rx}$ and their respective technically-incurred losses $L_{tx}$ and $L_{rx}$, it defines the *link budget*—the received power $P_{rx}$ usually expressed logarithmically in decibel per milliwatt (dBm).

$$P_{rx} = P_{tx} + G_{tx} + G_{rx} - L_{tx} - L_{rx} - L_p \tag{1}$$

Since all RHS parameters except $L_p$ are either known in advance due to the choice of the technical equipment (i.e. $G_{tx}$ and $L_{tx}$) or hardly observable (i.e. $G_{rx}$ and $L_{rx}$), I assume $G_{tx} + G_{rx} - L_{tx} - L_{rx} = 0$ in the following, leading to a simplified link budget defined as:

$$P_{rx} = P_{tx} - L_p \tag{2}$$

Intuitively, Eq 2 thus states that the signal strength observed on a MS solely depends on the output power of the connected antenna and the loss in signal strength that occurs along the way between antenna and MS. Given the abundance of available models, I follow the guidance of the European Conference of Postal and Telecommunications Administrations (CEPT) on radio propagation simulation for mobile services and opt for the widely popular extended HATA model [52], named after Masaharu Hata, the author of the 1980 landmark study on the "Empirical Formula for Propagation Loss in Land Mobile Radio Services" [53]. It is derived from the COST-231 HATA model [54], which in turn builds on the original HATA [53] and Okumura model [55]. They all have in common that they are empirical models to estimate the median path loss between a transmitter and a receiver based on real-world measurements. The HATA model extends the Okumura model by distinguishing between urban, suburban and rural settings, thus accounting for different levels of mean attenuation due to obstacles and changes in terrain. The COST-231 HATA model increases the frequency range of the original HATA model. The extended HATA model is applicable for settings with frequencies $f$ between 30-3000 MHz, distances $d$ between 0-100km, transmitter heights $h_{tx}$ between 30-200m and receiver heights $h_{rx}$ between 1-10m. The general form of the extended HATA model $L_p^{EH}$ consists of a loss function $L$ for the median path loss and a path loss variation term $V$ drawn from a log-normal distribution that accounts for the stochastic nature of radio propagation Since model parameters vary depending on the distance, the expected environment $env$ (indoor/outdoor and rural/suburban/urban) and the frequency, the full extended HATA model is not spelled out in this paper, but can be accessed here: https://ecocfl.cept.org/display/SH/A17.3.1 +Outdoor-outdoor+propagation.

$$L_p^{EH}(f, d, h_{tx}, h_{rx}, env) = L(f, d, h_{tx}, h_{rx}, env) + V(\mu, \sigma, d) \tag{3}$$

As an example, I provide the path loss function of the extended HATA model $L_p^{EH}$ for distances above 0.1km outdoor in rural areas for frequencies between 150 and 1500 MHz:

$$
\begin{aligned}
L_p^{EH} = 69.6+ \\
46.09 * \log_{10}f- \\
13.82 * \log_{10}h_{tx}+ \\
(44.9 - 6.55 * \log_{10}h_{tx}) * \log_{10}d- \\
(1.1 * \log_{10}f - 0.7) * h_{rx}- \\
20 * \log_{10}(h_{rx}/10)- \\
20 * \log_{10}(h_{tx}/30)- \\
4.78 * (\log_{10}f)^2- \\
40.14+ \\
V(12, 12)
\end{aligned}
\tag{4}
$$

So, for example, an MS 1m above the ground at a line-of-sight distance of 3km in a rural area to an omnidirectional antenna that is 30m above the ground transmitting at the 900 MHz frequency band would experience a path loss of $L_p^{EH} \approx 118 dBm$. Assuming a GSM macro-cell with an output power $P_{tx} = 43\ dBm$ using Eq 2 yields a budget for that link, also known as *received signal strength* (RSS), of $P_{rx} = P_{tx} - L_p^{EH} \approx -75 dBm$. As a rule of thumb, signals with RSS values above $-80\ dBm$ are considered excellent, RSS values below $-110\ dBm$ point to very poor signals.

## Methodology

Usually, statistical data on individuals or households are geo-located to statistical areas via their respective *places of residence*. Further, unit-level data is aggregated to area-level aggregates using some form of weighting factor such as survey weights. For example, the poverty rate of a region can either be calculated as the share of units classified as *poor* among the interviewed residents of the region multiplied by their sampling weight or via sub-regional poverty rates weighted with the respective sub-regional population counts. However, neither the places of residences nor the weights are generally available on the cell-level of a mobile network (as an equivalent to the sub-region). Hence, they need to be estimated.

In mobile phone metadata analysis, the place of residence of an individual/household is usually approximated with the night-time home location of an MS recorded at the cell-level.

To derive survey weight proxies, for example, point-to-polygon allocation assumes equal weights for all cells point-located within a statistical area. Voronoi tessellation uses the area size of the intersection of voronoi tile and statistical area as weighting factor, i.e. 1 $km^2$ always conveys the same importance in aggregation, no matter whether it is 1 $km^2$ of sparsely-inhabited desert or 1 $km^2$ of a densely-populated city.

In most cases, the place of residence of an individual/household (thus is approximation alike) is linked to some form of settlement. However, neither the statistical area nor the coverage area of a cell account for that fact. Consequently, the underlying idea behind the proposed methodology is to use human settlement information extracted from publicly available satellite imagery as common geographic reference level for both statistical units such as households and home-located MS. This allows to a) construct weights based on settlement counts and b) refine weights in cases where MS counts, often regarded as highly sensitive information by the MNO, are available. Further, in combination with technical information on the antenna, it allows for an efficient coverage estimation to address the issues of holes and overlaps in a mobile network.

In the following, settlements are denoted as *i*, BTS as *j*, statistical areas as *t*, the number of home-located MS as *d*, the population count as *p*, the number of settlements as *n* and metadata covariates as *R*. To illustrate the value added of the proposed methodologies, Fig 2a and Table 1 showcase a typical setup faced when one seeks to augment official statistics with mobile phone metadata: statistical indicators are provided for statistical areas A, B and C. Mobile phone metadata is provided as BTS-level aggregates with the corresponding point locations 1 and 2. To account for that, I treat each cell site that may host multiple antennas as single omnidirectional antenna, calling it *BTS* subsequently. This constitutes a simplification of real mobile networks where usually multiple directional antennas serving on various frequency bands are co-located at the same site that does not necessarily have to be an actual (cell) tower. Although accounting for directionality of antennas as done by e.g. [4] is likely to affect the overall outcome of later analysis by increasing the number of network tiles available for mapping, the challenges for allocating them correctly (holes, non-linearities, overlaps, within-cell

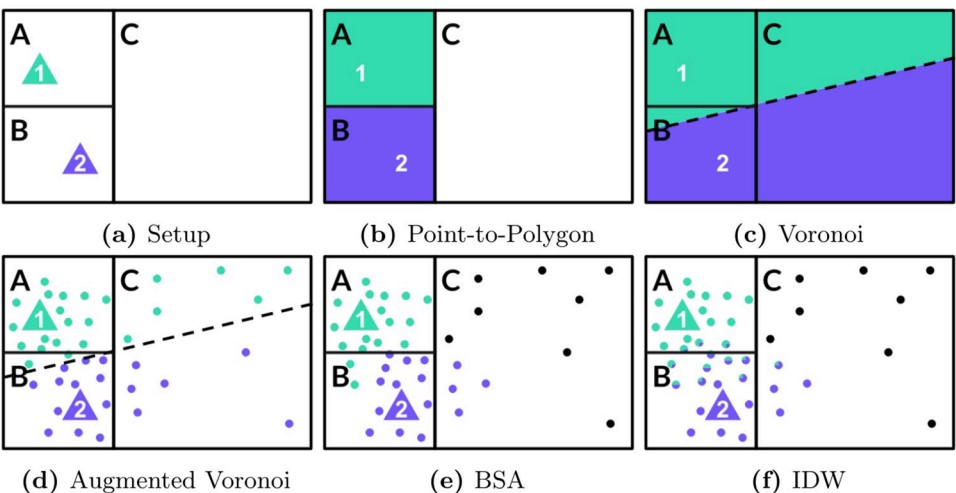

**Fig 2. Popular and proposed mapping schemes.** Three statistical areas (A-C), two BTS (1-2) and numerous dots representing built-up areas illustrate how different mapping schemes affect the allocation of BTS-level data to statistical data.

heterogeneity) remain. Consequently, it is expected that results from this study also apply to a setup based on directional antennas, thereby justifying the simplifying assumption. Further details on Fig 2b–2f are provided in the following subsections.

## Point-to-polygon allocation

For purposes such as model fitting one approach to combine statistical data and mobile phone metadata is to aggregate metadata covariates onto the same geographical level, e.g. statistical areas. To do so, the point-to-polygon approach (*p2p*) treats BTS point locations as such and allocates BTS-level metadata covariates using a binary weighting scheme (see Fig 2b and Eq 5).

$$w_{j,t}^{p2p} := \begin{cases} 1 & \text{if } j \subseteq t \\ 0 & \text{otherwise} \end{cases} \tag{5}$$

Consequently, all network traffic handled by a BTS is attributed to one statistical area exclusively, no matter whether it was generated by a home-located MS actually 'residing' in this area or not. In the toy example, but also in the real-world application presented in Section Application this leads to a situation where no metadata covariates are available for certain area, e.g. area C—with negative effects on the final sample size in model fitting.

## Voronoi tessellation

In contrast, voronoi tessellation (denoted by superscript *v*) divides the total space of interest into perfectly disjunct tiles along the equidistant lines between points, in this case the BTS

**Table 1. Example of statistical data and mobile phone metadata.**

| area_id | poverty_rate | bts_id | # of calls | lon | lat |
|---------|--------------|--------|------------|---------|----------|
| 1 | 0.23 | 6453 | 34050 | 43.2344 | 23.2342 |
| 2 | 0.11 | 8348 | 1023 | 50.0988 | 18.84217 |

point locations (see Fig 2c). The current state-of-the-art procedure is to intersect these tiles—representing approximated coverage areas of BTS—with the statistical areas. The weights to aggregate BTS-level metadata covariates to the respective statistical area are derived from the size of the intersection of tiles $a_j$ and $a_t$ of BTS $j$ and statistical area $t$, respectively, in relation to the total size of $a_t$, also expressed as

$$w_{j,t}^v := \frac{a_j \cap a_t}{a_t} \tag{6}$$

In the toy example of Fig 2c, this would reduce to be the intersection of e.g. statistical area A and the voronoi tile of BTS 1 divided by the total area of A. However, as mentioned above, area sizes are used in that approach to approximate the (usually) unknown population counts per intersection by implicitly assuming homogeneous distribution of the population within a given statistical area.

## Augmented voronoi tessellation

The proposed settlement-based mapping schemes relax this obviously strong assumption by assuming a homogeneous housing structure instead, i.e. a constant population density per settlement area within a given statistical area. Applied to voronoi tessellation, Fig 2c and 2d—with settlement areas represented as dots—illustrate the difference. Instead of using the area sizes $a_j$ and $a_t$ to calculate the weights, the "augmented" voronoi tessellation (*av*) uses the number of settlements per area, denoted as $n_j$ and $n_t$, respectively.

$$w_{j,t}^{av} := \frac{n_j \cap n_t}{n_t} \tag{7}$$

Consequently, statistical area-level covariates can easily be acquired for both approaches using a weighted average (or a weighted median) on BTS-level data.

$$\hat{R}_t = \sum_{j=1}^{J} w_{j,t} R_j \tag{8}$$

Going back to the toy example, while BTS 1 covers the smaller part of C in Fig 2c, thus receives a smaller weight in the calculation of area-level metadata aggregates, it looks different in Fig 2d when comparing the number of settlements, represented by green and purple dots. This way, the proposed methodology accounts for within-cell heterogeneity of the population distribution.

Both voronoi tessellation and augmented voronoi tessellation splits the full space of interest into disjunct tiles. Applied to a mobile network this means ubiquituous coverage and zero redundancies, i.e. all dots are uniquely associated to a specific BTS in the toy example. Again this is a strong assumption that most likely does not hold true in any real-world application. To relax this assumption by introducing holes and overlaps in the network coverage, additional information are necessary that allow for the estimation of coverage measures such as the received signal strength (RSS) at any given point in space. Fig 2e exemplifies the consequences: Some settlements are not covered (black dots) and some settlements, even though closer to one BTS, receive a stronger signal from a more distant BTS. Assuming coverages are correctly estimated in Fig 2e and 2f, it demonstrates that point-to-polygon allocation tends to underestimate the coverage of statistical areas while voronoi tessellation tends to overestimate it.

## Propagation-based mapping schemes

Previously presented schemes follow a 'BTS-centric' approach by first determining the respective coverage area of a BTS and then analyzing potential overlaps with other places of interest such as settlements. In contrast, propagation-based schemes follow an 'MS-centric' approach by looking at the connectivity at the place of interest, i.e. the place of usual residence or the home location first and then estimating which (group of) BTS it most likely serves. As outlined in Section Radio propagation modelling, multiple ways exist to estimate the 'connectivity' of an MS, but all require at least information on the distance to the surrounding BTS and additional technical specifications. With that, the serving BTS can be determined at each place of interest, thus allowing for a more nuanced coverage mapping. Here, settlements can provide a common geographic reference for the *place usual residence* and the *home location* alike.

**Best server area (BSA).**   In mobile networks, an MS usually connects to the antenna that offers the strongest signal. Thus, the settlement-level weight is 1 for the BTS with the strongest signal and 0 otherwise.

$$
w_{i,j}^{bsa} := \begin{cases} 1 & \text{if } P_{rx,i,j} = max(P_{rx,i,\cdot}) \\ 0 & \text{otherwise,} \end{cases}
\tag{9}
$$

Links weaker than a certain threshold (e.g. a $P_{rx}$ value below—110 dBm) can be discarded as they represent 'dead' links. This way the approach accounts for holes in the network coverage. The weights $w_{i,j}$ express the importance of a BTS for a pixel. Similarly to Eq 8, they can be used to determine the statistical area-level covariate estimates $\hat{R}_t$ using a weighted average:

$$
\hat{R}_t = \sum_{i=1}^{n_t} \frac{w_{i,j}}{\sum_{i=1}^{n_t} w_{i,j}} R_j
\tag{10}
$$

Due to the binary nature of the weight, $\sum_{i=1}^{n_t} w_{i,j}$ represents the number of settlements with mobile coverage within a given statistical area. In areas with homogeneous network infrastructure and full coverage, the best server approach closely resembles the augmented voronoi tessellation with the difference that path loss increases non-linearly with the distance, i.e. locations very close to the location of a BTS may be served by another, more distant one.

**Inverse signal strength.**   Radio propagation is stochastic by nature. Changing environmental conditions and varying network loads affect the RSS at a given location across time. Consequently, the strongest signal is not always provided by the same BTS. In order to assure quality of service, mobile networks usually exhibit a certain number of overlaps. To account for that, I calculate inverse distance weights (IDW) for each pixel $i$ using the median link budget $P_{rx,i,j}$ as non-linear distance measure (see Eq 11) to the k-nearest antennas. $s$ denotes a tuning parameter, where $s = 0$ reduces $w_{i,j}^{idw}$ to a fixed weight per BTS and a large $s$ can be used to approximate the best server approach.

$$
w_{i,j}^{idw} := \frac{v_{i,j}}{\sum_{j=1}^{k_i} v_{i,j}} \qquad \text{with } v_{i,j} := \frac{1}{|P_{rx,i,j}|^s} \qquad \forall j \in k_i
\tag{11}
$$

Here again, $w_{i,j}^{idw}$ can be used to calculate statistical area-level weighted averages of BTS-level mobile phone metadata covariates as presented in Eq 10.

## Potential extensions

Depending on data availability, the methodology can further be extended. While MNOs often regard MS counts as highly sensitive information since they reveal a detailed picture of local market shares, they can be used to further refine the weights towards more accurate population counts. [4] presents elaborate approaches to use MS counts and advanced technical network specifications to derive high-resolution population density estimates from signalling data.

Further, high-resolution population grid estimates such as provided by WorldPop at 100x100m [12] can be used as an alternative to binary settlement data. Here, $\hat{w}_{i,j}$ can be substituted with the estimated population count $\hat{p}_i$ per pixel directly extracted from the image.

## Simulation

In order to evaluate the underlying motivation behind this methodology, i.e. more accurate mapping schemes produce more accurate outcomes, I test the performance of the different mapping approaches in terms of their overlap with the true coverage area and the accuracy of the predictions in a controlled setting with groundtruth information. Therefore, I run a simulation $T = 1000$ times on a synthetic population grid in which I re-distribute individuals, their poverty status, BTS locations and technical BTS specifications randomly. I observe the geographical overlap of the true and the estimated coverage areas, the overlap in home-located settlements and the correlation between the true and the estimated variable of interest (in this case the *poverty rate*). The main challenge in this simulation is to create "true" coverage areas for each BTS that provide a realistic, but simplified benchmark for this study. Consequently, I opt for the extended HATA model. The choice is motivated by a series of propagation model evaluations using real-world measurements, notably [56–58]. The stochastic component within the HATA model is disabled in order to isolate the effect of interest.

### Setup

I simulate a country including a major city, an uninhabited area such as a large lake or a national park and rural area otherwise using a 1000 x 1000 grid where each quadratic pixel represents an edge length of 100m. The urban area is divided into 16 equally-sized (50 x 50 pixel) small statistical areas, whereas the rural area is divided into 24 larger ones (200 x 200 pixel). I randomly distribute one million individuals across the grid using a multivariate normal distributions with $\mu_x = 10$, $\mu_y = 10$, $\Sigma_x = [50, 0]$ and $\Sigma_y = [0, 50]$ for the urban area (1/2 of the total population) and varying parameter values for the rural centers and a uniform distribution for the remaining rural area. Pixel-level population counts are calculated from individual-level data. Fig 3 shows an example of the settlement distribution across space and the corresponding population density.

In the next step, I randomly assign a poverty rate to each pixel. First, I generate a 4x4-pixel poverty grid for which I calculate the population density (see Fig 4b). In order to account for differences in the poverty rate between urban and rural areas, I randomly draw from a uniform distribution with values between 0 and 1 and multiply it with the inverted normalized population density. This poverty rate serves as the mean $\mu$ for randomly assigning poverty rates to settlements within the respective grid area using a normal distribution $N(\mu, \sigma)$ with $\sigma = 0.5$. Values below 0 and above 1 are windsorized. This two-step procedure tries to limit good predictive performances for areas not actually covered due to inference facilitated by the same underlying data generating process. Further, I assume that every inhabitant has one and only one MS and that there exists an indicator derived from mobile phone metadata that perfectly correlates with the true poverty rate of a given set of MS. Consequently, deviations in the

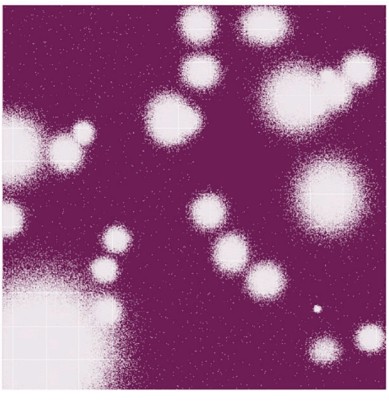
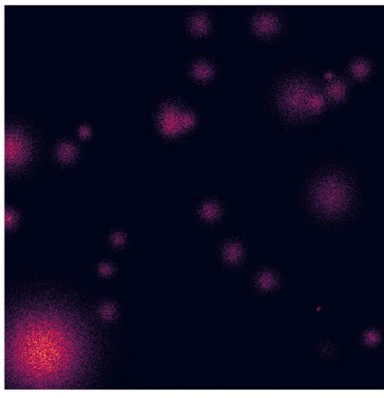

**(a) Settlements**                    **(b) Population density**

**Fig 3. Simulation setup—Settlements.** (a) shows locations of the built-up areas in a hypothetical country, while (b) shows the corresponding population density in these areas (the brighter the colour, the higher the population density).

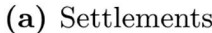

correlation between the poverty rate captured via the "true" coverage area and the poverty rate captured via the estimated coverage area exclusively originate in their coverage mismatch.

In order to create a mobile network on top of that structure, I use a clustering algorithm based on the population density (see Fig 5b). BTS are distributed across the country at a ratio of roughly 1 BTS per 5,000 inhabitants in urban areas and 1 BTS per 10,000 inhabitants in rural areas. This results in 100 urban and 50 rural BTS in this simulation. BTS are interpreted as omnidirectional antennas and assigned specific heights, frequencies and output powers. The specifications vary more strongly in the urban area in order to reflect the greater complexity of network topology generally found in metropolitan areas. Since the HATA model requires a classification of areas into *urban*, *suburban* and *rural*, I use those 50% of BTS with the smallest number of pixels associated to them by the clustering algorithm used above as *urban* and those 5% of BTS with the largest number of pixels as *rural*, *suburban* otherwise. At the end, BTS heights are between 15—60 m with frequencies at 900 MHz and 2100 MHz and output power between 40 and 47 dBm. The MS height is fixed at 1m above ground level.

Based on these technical specifications, the true coverage areas and the true home locations of the settlements using the extended HATA model are calculated and used to create benchmark estimates of the true poverty rate. The results are then compared against estimates from point-to-polygon allocation, voronoi tessellation, augmented voronoi tessellation and BSA and IDW approaches of a naïve ('simple') version of the extended HATA model that does not know the exact technical BTS specifications, but makes an educated guess based on publicly available information such as the frequencies used in the country and the location of urban centers. Fig 6 exemplifies how the approaches differ in terms of geographical coverage.

The results are compared in three different ways: How much do they overlap geographically? How much do they overlap in terms of home-located settlements? How well do they predict the true poverty rate of a given statistical area?

## Results

Table 2 shows the best performing approach in each round across round for all five performance indicators. Performance differences between voronoi tessellation versus the augmented voronoi tessellation and the augmented voronoi tessellation versus the HATA (BSA) approach

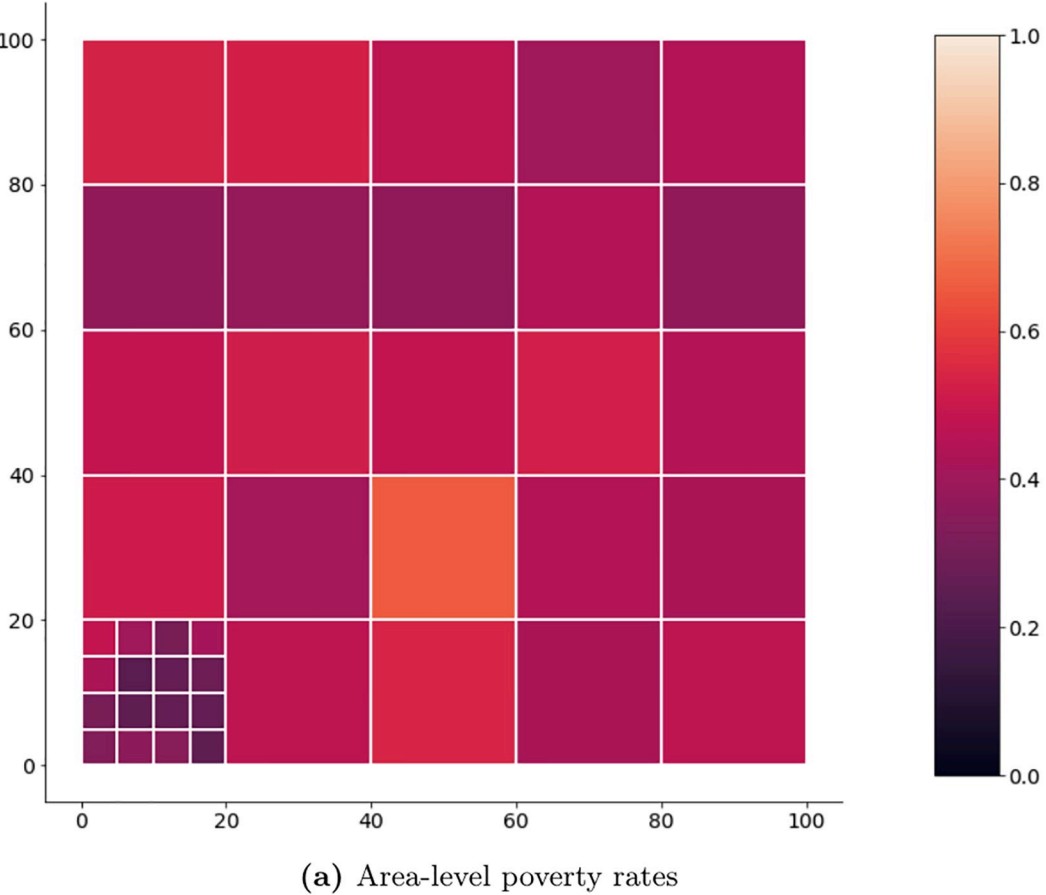

(a) Area-level poverty rates

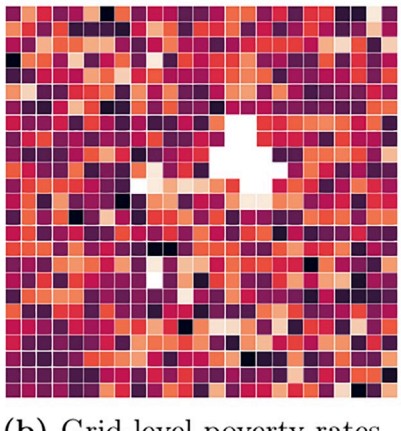

(b) Grid-level poverty rates

(c) Settlement-level poverty rates

**Fig 4. Simulation setup—True poverty rate.**

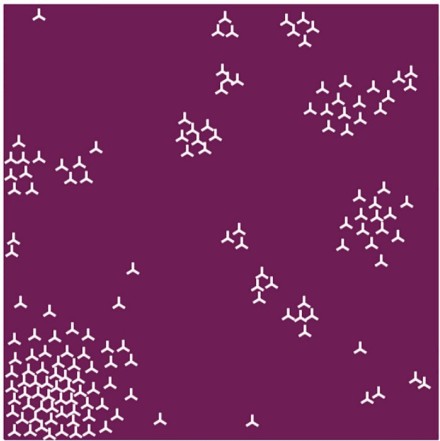

**(a)** BTS locations        **(b)** BTS locations and settlements

**Fig 5. Simulation setup—BTS locations.**

showcase the relative contribution of settlement weighting and radio-propagation modelling, respectively. As expected, the simple HATA model clearly outperforms the other mapping approaches in terms of overlap, both geographically with the true coverage area (see Table 3) as well as concerning the home-located settlements (see Table 4). As the settlement-based approaches do only affect the calculation of weights and not of the coverage area, the coverage results are identical for voronoi tessellation and augmented voronoi tessellation and for the two HATA approaches, respectively. However, this advantage is not reflected to a similar extent in the predictive performance.

Interestingly, the HATA (IDW) approach performs poorly in prediction in contrast to the HATA (BSA) approach. This is due to the fact that the poverty rate in the true coverage area is calculated based on a deterministic home location, i.e. it is calculated from a constant set of settlements. This coincides directly with the mode-based HATA (BSA) approach, however, it does not reflect most real-world settings, in which stochastic radio propagation and overlapping coverage areas lead to situations where the captured poverty rate by the BTS is sourced from varying sets of settlements. The HATA (IDW) approach addresses this setup. Consequently, it is expected that the differences between these two approaches at least diminish in the application with real-world data in Section Application. Also, deviations of the HATA (BSA) approach from the benchmark exclusively originate in the technical misspecifications as the true coverage area is calculated from a correctly specified HATA model. The network complexity faced in real-world settings is expected to further undermine the accuracy of propagation-based mapping schemes.

Looking at the performance of the two voronoi approaches in Table 2 the value added of using settlement information becomes apparent. Recalling the setup, the simulation assumes error-free human settlement identification. This, again, may not hold true in a real-world application as some buildings may not be detected while some detected buildings may not be inhabited. Consequently, it is expected that the difference between thee two voronoi approaches will be less stark in the application.

Fig 7 shows the distribution of the three performance indicators across rounds for those statistical areas for which every mapping scheme can provide estimates. On average, this reduces the underlying set of observations from 40 to 32 (see the sample sizes in Table 5). The result

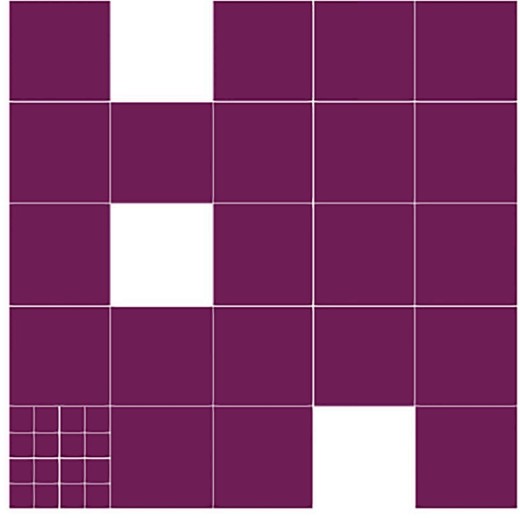

**(a)** Point-to-Polygon

**(b)** Voronoi tessellation

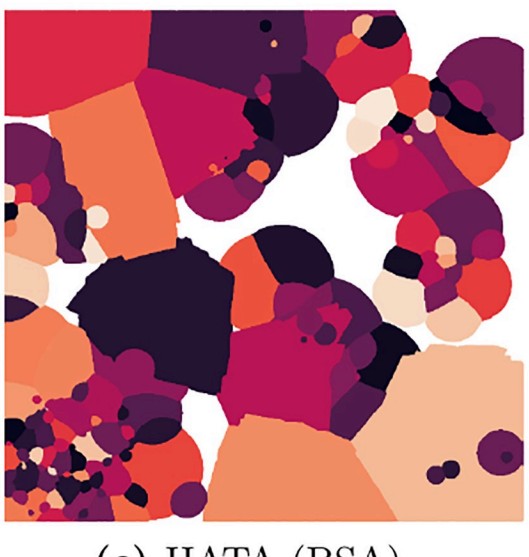

**(c)** HATA (BSA)

**(d)** HATA (IDW)

**Fig 6. Coverage areas exemplified.**

for the true coverage area are represented as *benchmark* for the other approaches as it estimates the settlement-level poverty rates actually captured by the respective BTS. Consequently, the benchmark should provide the upper bound for the $R^2$ and the lower bound for the bias and the RMSE in each round. Deviations thereof may only be due to spurious correlation.

The sample size difference also explains the difference between the performance of the point-to-polygon approach in terms of correlation in Table 5 vis-à-vis the performance metrics, especially in rural areas. Point-to-polygon allocation does not provide poverty estimates for 8 out of 40 statistical areas, on average, as they do not host a BTS (cf. Fig 6b). As both

**Table 2. Best performing approach by round across rounds (in%).**

| Mapping | Coverage | | Prediction | | |
|---|---|---|---|---|---|
| | Geography | Settlements | $R^2$ | Bias | RMSE |
| Point | 0.0 | 0.0 | 27.5 | 28.2 | 28.6 |
| Voronoi | 0.0 | 0.07 | 2.6 | 9.9 | 2.1 |
| Aug. Voronoi (GUF) | | | 35.5 | 33.1 | 36.8 |
| HATA (GUF, BSA) | 100.0 | 99.3 | 29.7 | 13.7 | 27.7 |
| HATA (GUF, IDW) | | | 4.7 | 15.1 | 4.8 |

poverty rate and BTS allocation is linked to the population density by design, it can be expected that the predictive performance for rural areas not hosting a BTS are poor as they are generated from different underlying distributions.

However, this does not fully explain the performance differences between the approaches. On one hand, statistical areas are quite large, thus most of the BTS experience little overlaps in their true coverage area with other statistical areas. Consequently, the statistical area provides a decent approximation for the coverage. In contrast, simple voronoi tessellation with geographical weights tends to overemphasize the importance of remote areas as a) it assumes to cover areas for which data is actually not captured and b) BTS are usually located in close proximity to populated areas while serving remote areas further away as a side effect of it. This may be especially relevant in situations with large between-variation among statistical areas, strong population clusters and imperfect mobile network coverage. While b) is accounted for in the simulation, only approx. 0.1% of the settlements are not covered by the network. Although this in line with the mobile network coverage in most countries, it can be expected that propagation-based schemes that account for holes in the mobile network outperform established approaches in setups with poor coverage.

## Application

In their 2017 study on estimating literacy rates in Senegal published in the *Journal of the Royal Statistical Society Series A*, Schmid et al. [1] use point-to-polygon allocation to map BTS point locations to statistical areas (*communes*). I revisit the design-based simulation of the study and extend it with four alternative mapping schemes, notably voronoi tessellation, satellite-augmented voronoi tessellation and the herein presented propagation-based coverage estimation methods using the best server area approach and the inverse signal strength weights. I compare the outcomes of all five schemes in terms of bias, root mean squared error (RMSE) and adjusted $R^2$.

### Situation in Senegal

The application draws on real-world data from Orange-Sonatel for the year of 2013 [7]. During that time, the MNO operated mainly on the GSM 900 (2G) band with some UMTS 2100 (3G) deployments in urban centers. A large share of on-net traffic (approx. 91% of overall

**Table 3. Geographical overlap with true coverage area (in%).**

| Mapping | Total | Rural | Suburban | Urban |
|---|---|---|---|---|
| Point | 25.8 | 15.3 | 30.9 | 22.3 |
| Voronoi | 30.7 | 14.1 | 25.5 | 37.0 |
| Simple HATA | 55.3 | 80.1 | 62.1 | 46.7 |

**Table 4. Overlap with true home-located settlements (in%).**

| Mapping | Total | Rural | Suburban | Urban |
|---|---|---|---|---|
| Point | 16.9 | 44.3 | 15.9 | 14.9 |
| Voronoi | 54.2 | 56.8 | 60.7 | 48.0 |
| Simple HATA | 59.7 | 87.6 | 66.5 | 50.6 |

traffic vis-à-vis a market share of approx. 57%) during that year suggests a high prevalence of dual SIM use. It is expected that in this setting a negligible share of SIM cards are used by IoT devices others than MS. Coverage advantages in rural areas suggest dual-SIM use to be a phenomenon of more densely populated areas. The country exhibits little irregularities in the terrain: The highest point of Senegal being approx. 648 m above sea level is located at its southern border. The lowest point constitutes the sea level. Urban built-up areas with multi-storey buildings are predominantly limited to downtown Dakar. Most of the country is dominated by savanna with sparse high-grown vegetation.

## Original study

In their design-based simulation, Schmid et al. [1] implement a stratified two-stage cluster sample design similar to the one used in large-scale household surveys such as the Demographic and Health Survey (DHS) using a 10% random sample of a pseudo-population as sampling frame, the 431 communes of Senegal as primary sampling units (PSUs) and the 14 regions of Senegal as strata. The authors combine the constructed 'survey' data with covariates extracted from mobile phone metadata on the level of communes in order to evaluate different small area estimation techniques using the *unemployment rate* as target variable of choice. The 72 available covariates are calculated on the subscriber-level using the Python library *Bandicoot* [8]. The subscriber-level covariates are allocated and aggregated to a BTS using the most frequently used BTS by a subscriber between 7pm and 7am as the *home location*. The BTS-level covariates are then allocated and aggregated using point-in-polygon allocation. Variable selection is performed backwards on large communes using the Bayesian Information Criterion. The covariates are used to generate small area unemployment rate estimates using a transformed Fay-Herriot model. Finally, Schmid et al. evaluate the small area estimates against the 'true' pseudo-population aggregates in 500 simulation runs using bias and RSME for a) communes covered by the survey (*in-sample*) b) communes not covered by the survey (*out-of-sample*) and c) communes without covariates from mobile phone metadata. For additional details on the setup of the original study, I refer to [1].

## Extensions

I re-run the simulation of the original study five times thereby only varying the commune-level matrix of covariates as inputs. Specifically, I create five distinct sets of commune-level covariates beforehand by applying different mapping schemes during the aggregation process of the BTS-level data of the original study. First, I use the point-to-polygon allocation used in the original study. Second, I apply a standard voronoi tessellation to extract spatial weights proportional to the geographical overlap of tile and statistical area as described in Voronoi tessellation since it is used in most other studies in this field. Third, I augment the voronoi tessellation with settlement information from GUF by taking the number of white pixels (representing (part of) a settlement) within each section as a weight for commune-level aggregates to account for within-cell heterogeneity. Fourth, I implement the extended HATA (BSA)

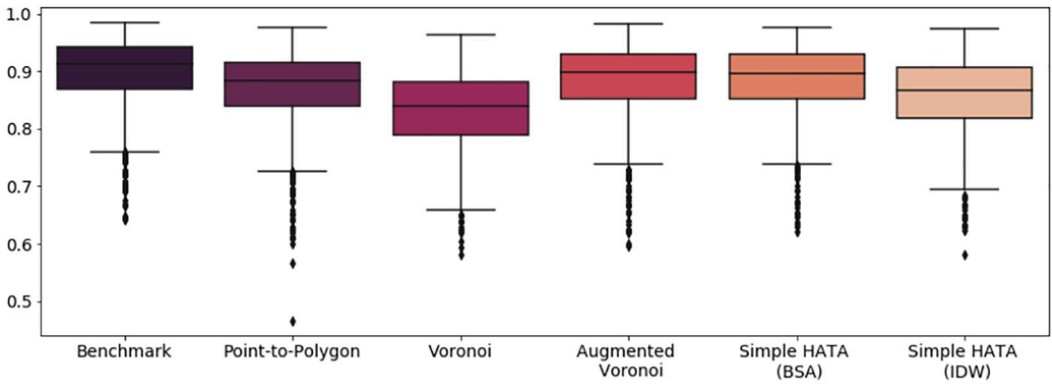

(a) $R^2$

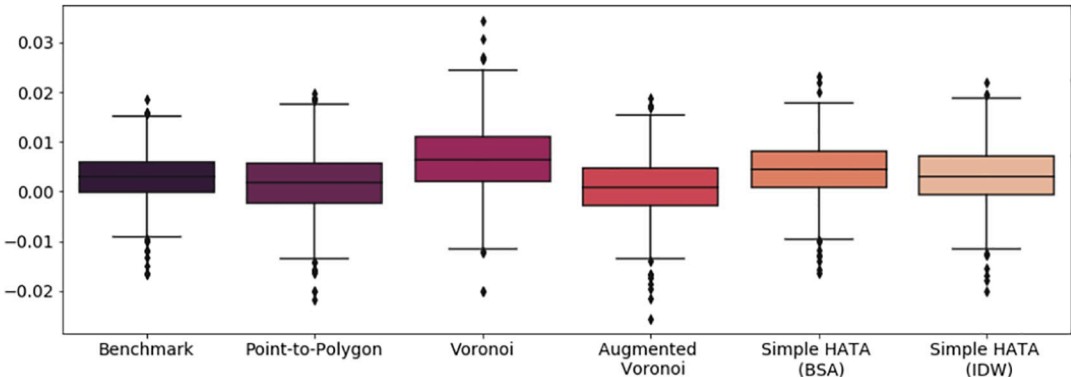

(b) Bias

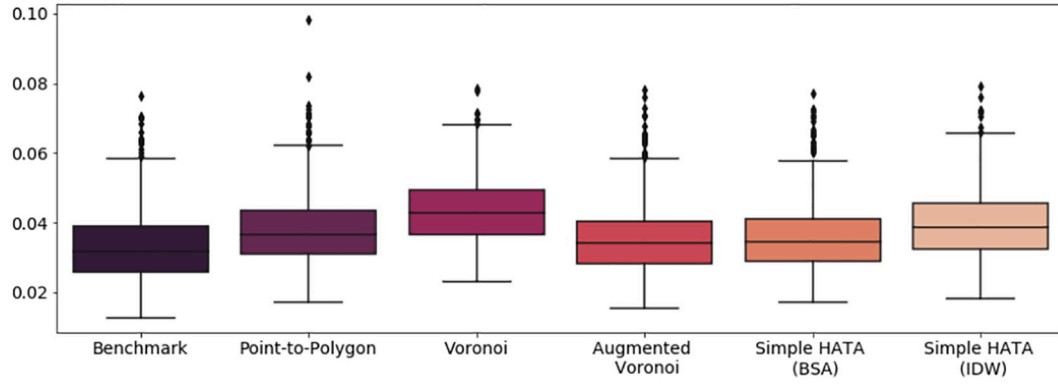

(c) RMSE

**Fig 7. Estimating the true poverty rate for statistical areas.** Distribution of the three performance metrics adjusted $R^2$, bias and RMSE with the estimated poverty rate using the true coverage area, i.e. built-up areas perfectly allocated to BTS, as 'Benchmark' across 1000 simulation runs.

**Table 5. Area-level correlation of estimated and true poverty rate & sample size.**

| Mapping | $\rho$ | $n$ | $\rho_{Rural}$ | $n_{Rural}$ | $\rho_{Urban}$ | $n_{Urban}$ |
|---|---|---|---|---|---|---|
| Benchmark | 0.905 | 40 | 0.734 | 24 | 0.971 | 16 |
| Point | 0.930 | 36 | 0.828 | 20 | 0.940 | 16 |
| Voronoi | 0.873 | 40 | 0.622 | 24 | 0.966 | 16 |
| Aug. Voronoi | 0.896 | 40 | 0.715 | 24 | 0.966 | 16 |
| Simple HATA (BSA) | 0.897 | 40 | 0.717 | 24 | 0.957 | 16 |
| Simple HATA (IDW) | 0.885 | 40 | 0.670 | 24 | 0.962 | 16 |

model as presented in Methodology and GUF data. In densely populated areas, this approach closely resembles voronoi tessellation, however, it allows for holes in the network and for non-linear relationships between signal strength and distance. Fifth, I use inverse signal strength weights—HATA (IDW)—to capture the stochastic nature of a link.

Comparing Fig 8c and 8d to the direct estimator (Fig 8b) shows the benefits of augmenting survey data with mobile phone metadata: providing estimates for small areas not originally covered by the survey. Looking at settlements in Fig 8a, it is noteworthy that one commune—Thietty in the region Kolda—does not appear to host any settlement identified as such in GUF data. While official population numbers do not support this view, it underlines the fact that information extracted from satellite imagery, e.g. settlement classifications, are subject to some degree of uncertainty.

## Assumptions

In contrast to point-to-polygon allocation and voronoi tessellation, the extended HATA model requires additional technical antenna specifications, notably the antenna and receiver height, the frequency and the transmitter power. As additional information are not available in the original study, I make following assumptions: I fix both the antenna height $h_{tx}$ and the receiver height $h_{rx}$ at the lower bound of the extended HATA model, which is 30 m and 1 m, respectively, both located outdoors with line-of-sight and a transceiver installed above the roof. As most of Senegal is flat without high multi-storey buildings except in downtown Dakar and in large parts no high-grown vegetation this assumption appears reasonable. Further, I fix the frequency in rural areas at 900 MHz and in urban centers at 2100 MHz and I interpret BTS as omnidirectional antennas with an output power of 45 dBm. This is clearly a simplification of the actual network topology, especially in urban areas with a mix of directed micro and macro cells. However, in Senegal in 2013, 4G has not yet been introduced and Orange-Sonatel was operating 3G (on the 2100 MHz frequency band) only in urban areas. The remaining country was served with 2G technology on the 900 MHz band. Comparing own estimates with coverage area estimates for 2G in 2017 published by Sonatel [59] allows for a rough sanity check for the assumptions.

While Senegal offers an official classification of *rural* and *urban* on the commune-level, it is imperfect for the purposes of this study, as it takes a wide variety of non-network-specific factors into account. This leads to a situation where places with a high population density, e.g. Touba Mosque, are classified as *commune rurale*. Instead, I use BTS density per $km^2$ as a proxy for urbanity with a threshold of 1. Communes with more than one BTS per $km^2$ are classified as *urban*, those 50% of the communes with the lowest site density are classified as *rural*, the remaining communes are classified as *suburban*. This represents a more network-oriented measure of urbanity and is also in line with the area type classification of the HATA model.

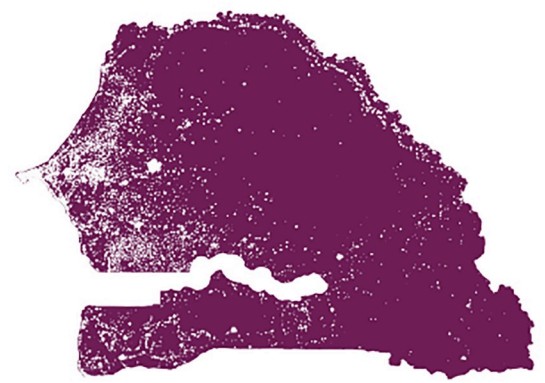

**(a)** Settlements

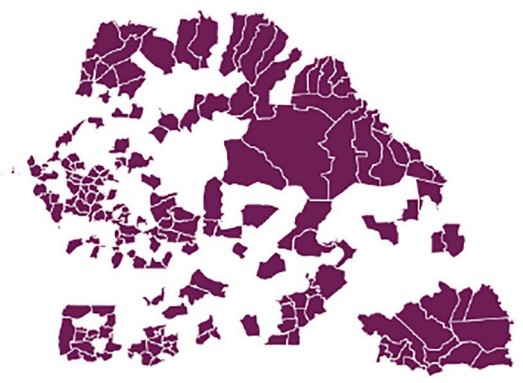

**(b)** Direct estimate

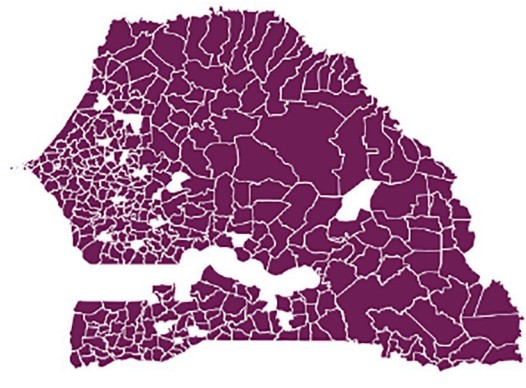

**(c)** Point-to-Polygon

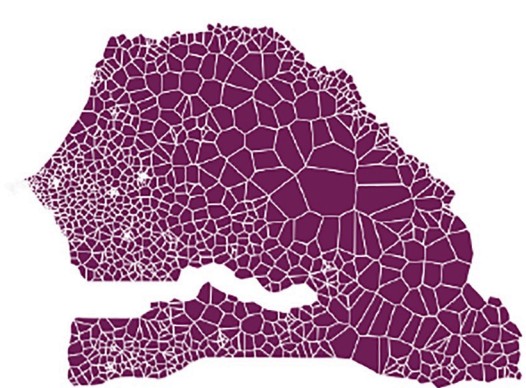

**(d)** Voronoi

**Fig 8. Commune-level coverage areas in Senegal.** Areas for which estimates of indicators of interest are available are coloured in red. Lower resolution built-settlements extents data reprinted from [10] under a CC BY license, with permission from WorldPop, original copyright 2018, are used in (a) for illustrative purposes.

## Results

Similar to Table 2 in the simulation, Table 6 shows which mapping scheme performed best across the 500 evaluation rounds. Confirming initial findings of Section Simulation, there is no clear winner. While point-to-polygon allocation performs best in out-of-sample predictions in terms of RMSE (54.0% of the rounds), it performs poorest in in-sample predictions. One possible explanation is that the lower average number of predictors used across rounds reduces the effects of overfitting. While HATA (IDW), HATA (BSA) and the augmented voronoi

**Table 6. Best performing approach by round across rounds (in%).**

| Mapping | Adj. $R^2$ | Bias | | RMSE | | Avg. # of predictors |
|---|---|---|---|---|---|---|
| | in | in | out | in | out | |
| Point | 6.0 | 16.4 | 23.2 | 12.6 | 54.0 | 4.2 |
| Voronoi | 10.2 | 21.0 | 16.6 | 21.6 | 22.8 | 5.0 |
| Aug. Voronoi (GUF) | 27.2 | 22.6 | 18.0 | 33.2 | 5.2 | 6.5 |
| HATA (GUF, BSA) | 27.0 | 21.8 | 16.4 | 18.0 | 7.8 | 6.4 |
| HATA (GUF, IDW) | 29.6 | 18.2 | 25.8 | 14.6 | 10.2 | 6.2 |

approach perform well across performance metrics, the overall difference between the approaches is limited (see Fig 9 and Table 7).

In contrast, urban communes do not perform significantly better than rural ones as suggested by the simulation results. Table 8 shows, similar to Table 5 for the simulation, the correlation between the actual and predicted commune-level unemployment rates. Fig 10 shows an orientation along the diagonal signalling overall good fit. A possible explanation is that the structural relationship of mobile phone metadata covariates and the unemployment rate is captured more robustly for rural areas as they constitute 385 out of 431 communes in Senegal. To test this explanation, Tables 1 and 2 in S1 Appendix show the results for in-sample and out-of-sample predictions by commune status, respectively. While urban communes outperform rural ones in in-sample prediction they fare worse for in the out-of-sample setting, thus supporting the aforementioned hypothesis.

While settlement-based mapping schemes exhibit improvements in the model fit compared to point allocation or voronoi tessellation, they do not translate into major efficiency gains in terms of bias and rmse (see Fig 9b and 9c). Possible reasons are threefold: There is a significant classification error in the settlement data. The complete absence of settlements in Thietty, Kolda, support this assumption. As a cross-check, I re-run the analysis with an alternative source of settlement information. Specifically, I use high-resolution population density estimates from WorldPop [12], however, it does not lead to gains in efficiency (cf. Table 3 in S1 Appendix). Second, there is high spatial auto-correlation, thus little structural difference between the densely and sparsely populated areas in terms of the variable of interest—here unemployment—so even though latter are overemphasized in the calculations, it does not affect the outcome predictions. Here, I re-run the application with alternative variables of interest, i.e. the *literacy rate* and the population count (cf. Tables 4 and 5 in S1 Appendix); again, without significant efficiency gains versa point allocation and voronoi tessellation. Third, there is little within-area variation of the population density so that geographic weights and settlement-based weights are very similar. The correlation coefficient between the weights of the two voronoi approaches confirm that with $\rho = 0.98$. Also, I use the 100 meters x 100 meters population estimates from WorldPop to extract commune-specific variation coefficients. For 76.8% of the communes, the within-commune variance is below 1, for 4% it is above 100 with a maximum at 3553.4.

In general, the value added of using propagation-based mapping schemes appears to be negligible in this application, even though official coverage area estimates by Sonatel [59] hint at the abundant presence of both overlaps and holes in the mobile network. A potential explanation is that the simplified HATA model is misspecified to an extent where the introduced errors cancel out the potential benefits. Looking at the specifications used in the application, this is most likely due to an underestimation of the coverage as the augmented voronoi

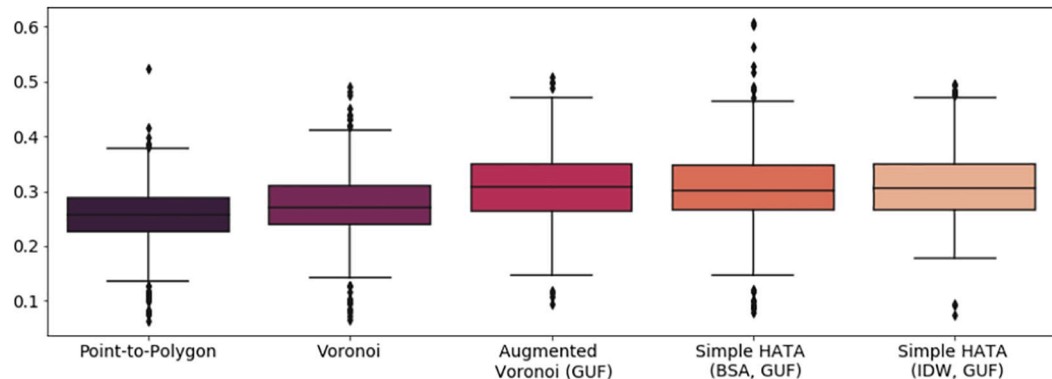

(a) Adjusted $R^2$

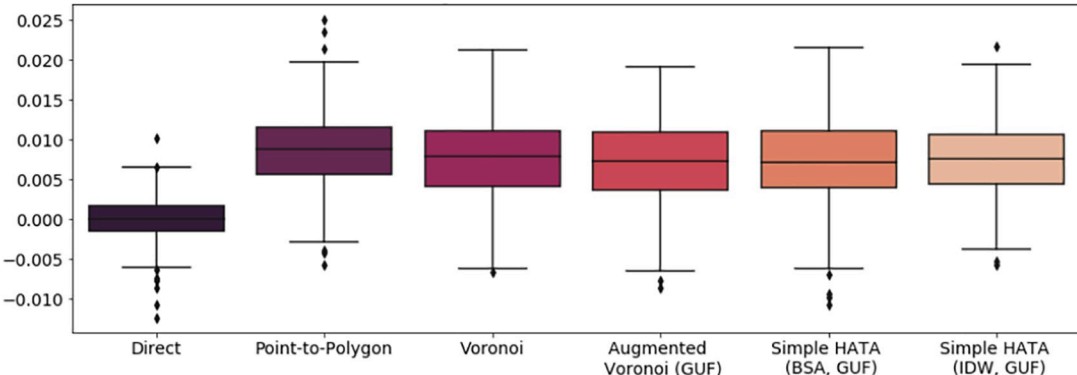

(b) Bias

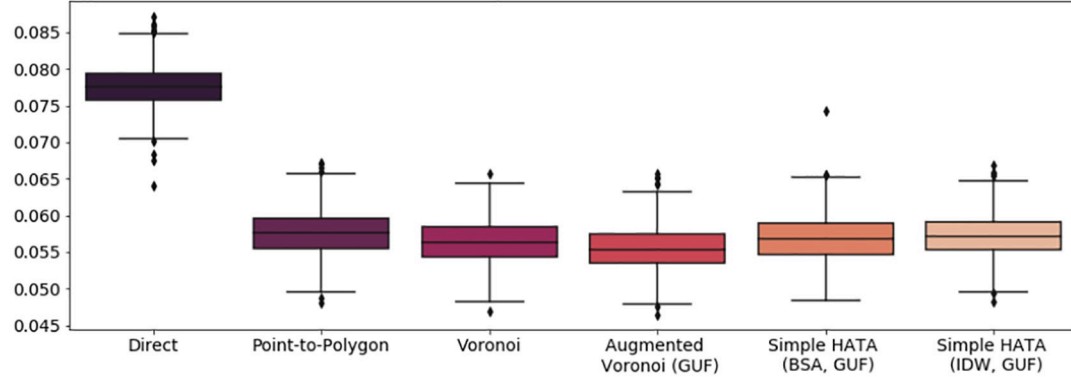

(c) RMSE

**Fig 9. Evaluation of poverty rate estimates for in-sample communes.** Distribution of the three performance metrics adjusted $R^2$, bias and RMSE across 500 simulation runs on a comparable set of communes. The typical trade-off between the bias and the variance of a small area estimator vis-à-vis the direct survey estimator becomes apparent.

**Table 7. Correlation with true unemployment rate and sample size in Senegal.**

| Mapping | $\rho$ | $n$ | $\rho_{in}$ | $n_{in}$ | $\rho_{out}$ | $n_{out}$ | $\rho_{ooc}$ | $n_{ooc}$ |
|---|---|---|---|---|---|---|---|---|
| Point | 0.535 | 431 | 0.765 | 192 | 0.320 | 210 | 0.355 | 29 |
| Voronoi | 0.542 | 431 | 0.778 | 196 | 0.313 | 235 | - | 0 |
| Aug. Voronoi (GUF) | 0.519 | 431 | 0.780 | 195 | 0.280 | 233 | 0.586 | 3 |
| HATA (GUF, BSA) | 0.511 | 431 | 0.770 | 194 | 0.269 | 232 | 0.670 | 5 |
| HATA (GUF, IDW) | 0.527 | 431 | 0.781 | 196 | 0.308 | 234 | - | 1 |

approach closely resembles the upper bound for an overestimation using the HATA (BSA) within a—by assumption—largely homogeneous network.

## Conclusion

Augmenting official statistics with mobile phone metadata still faces multiple methodological challenges, one of them is finding a common reference unit. As record-linkage on the individual-level presents considerable privacy risks a common procedure is to combine aggregates of these two disparate data sources on a geographical level. However, the stochastic nature of radio propagation makes it difficult to pin down coverage areas of the mobile network. Based on this study the good news is that it does not have to be complicated if supervised learning / prediction is the goal. While propagation-based models can help to refine the accuracy of coverage area estimation, it does not greatly impact the quality of the outcome predictions. One reason is that usually cells are located in a way that they provide a good service to as many MS as possible. As radio signals fade over distance, this means they are in close proximity to areas with high demand, i.e. densely populated places. Mapping schemes, in turn, mainly differ from each other when looking at the limits of a cell. However, most of the traffic which is correlated with statistical data for training/prediction is generated nearby, so the differences between mapping schemes become less relevant. Also, while geographical weights as used in most applications in this field ignore heterogeneity occurring within the cells, the corresponding statistical areas are often significantly larger. Therefore, cross-border cells, which could actually profit from weighting schemes that take within-cell heterogeneity into account, occur less frequent. In addition, cells and administrative (thus often statistical) areas are intimately linked via population clusters as both tend to be centered around them.

However, this study just provided initial evidence to inform future mapping choices and could be extended in multiple ways: First, both in the simulation and the application directional antennas are combined to omnidirectional antennas. While this is motivated by the typical data availability in real-world applications, it is of course a strong simplification of the actual network topology. As the lower bound of spatial heterogeneity captured is given by the number of unique areas resulting from intersecting coverage areas and statistical areas, studies

**Table 8. Area-level correlation of estimated and true unemployment rate & sample size.**

| Mapping | $\rho$ | $n$ | $\rho_{Rural}$ | $n_{Rural}$ | $\rho_{Urban}$ | $n_{Urban}$ |
|---|---|---|---|---|---|---|
| Point | 0.535 | 431 | 0.507 | 385 | 0.527 | 46 |
| Voronoi | 0.542 | 431 | 0.519 | 385 | 0.469 | 46 |
| Aug. Voronoi | 0.519 | 431 | 0.495 | 385 | 0.411 | 46 |
| Simple HATA (BSA) | 0.511 | 431 | 0.487 | 385 | 0.374 | 46 |
| Simple HATA (IDW) | 0.527 | 431 | 0.510 | 385 | 0.369 | 46 |

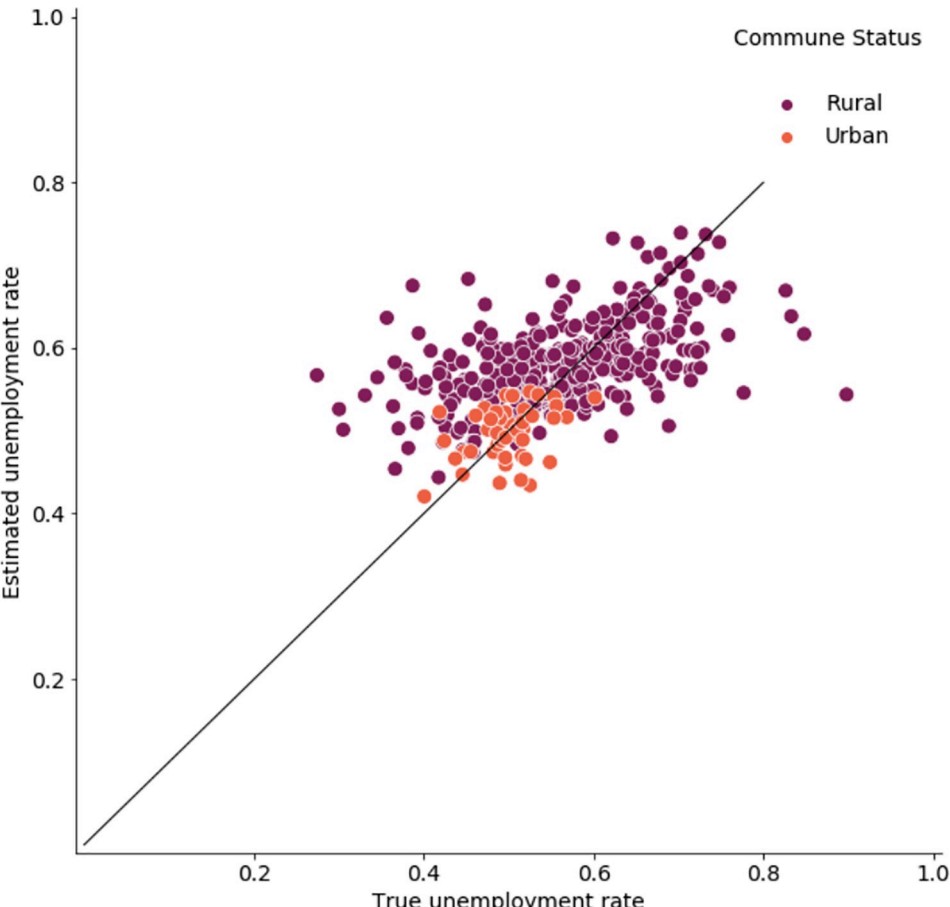

**Fig 10. True vs. estimated unemployment rate by commune status for a single simulation run.**

such as [4] have shown that moving from an BTS-oriented to a cell-oriented analysis could greatly affect analysis, especially via potential increases in sample size. However, it needs further investigation how refined mapping schemes can add further value, particularly in the presence of measurement uncertainty, to supervised learning setups in cell-level analysis. Second, the study used comparatively simple empirical propagation models based on real-world measurements largely ignoring actual environments. More advanced propagation models exist, however, they require significantly more computing resources that could limit their applicability as they take the physical surrounding via digital surface models into account. Nevertheless, investigating this constitutes an interesting path for further research.

## Supporting information

**S1 Appendix. Results and instructions.** Results from the cross-checks of the application and instructions for replicating the findings of this study.
(PDF)

**S1 File. Simulation.** Code for replicating the simulation study.
(ZIP)

**S2 File. Application.** Code and data for replicating the application study. See S1 Appendix for further details.
(ZIP)

## Acknowledgments

The author would like to thank Damien Jacques, Emmanuel Letouzé, Edward Oughton, Sören Pannier, Neeti Pokhriyal and Timo Schmid for excellent comments and helpful discussions.

## Author Contributions

**Conceptualization:** Till Koebe.

**Data curation:** Till Koebe.

**Formal analysis:** Till Koebe.

**Funding acquisition:** Till Koebe.

**Investigation:** Till Koebe.

**Methodology:** Till Koebe.

**Project administration:** Till Koebe.

**Resources:** Till Koebe.

**Software:** Till Koebe.

**Supervision:** Till Koebe.

**Validation:** Till Koebe.

**Visualization:** Till Koebe.

**Writing – original draft:** Till Koebe.

**Writing – review & editing:** Till Koebe.

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
