## [Decision Letter · Decision Letter 0]

15 Jul 2020

PONE-D-20-05427

Better coverage, better outcomes? Mapping mobile network data to official statistics using satellite imagery and radio propagation modelling

PLOS ONE

Dear Dr. Koebe,

Thank you for submitting your manuscript to PLOS ONE. After careful consideration, we feel that it has merit but does not fully meet PLOS ONE’s publication criteria as it currently stands. Therefore, we invite you to submit a revised version of the manuscript that addresses the points raised during the review process.

The paper should be corrected regarding the comments provided by the reviewers. Reviewer 1 provided major suggestions that may affect the substance of the paper's finding, but will certainly contribute to improve its quality and contributions. Also, make sure that all data underlying the findings described in the manuscript are fully available without restriction, as this is a requirement of the journal.

We look forward to receiving your revised manuscript.

Kind regards,

Jacinto Estima

Academic Editor

PLOS ONE

Journal Requirements:

2. We note that Figures 1, 8, 9 in your submission contain map images which may be copyrighted. All PLOS content is published under the Creative Commons Attribution License (CC BY 4.0), which means that the manuscript, images, and Supporting Information files will be freely available online, and any third party is permitted to access, download, copy, distribute, and use these materials in any way, even commercially, with proper attribution. For these reasons, we cannot publish previously copyrighted maps or satellite images created using proprietary data, such as Google software (Google Maps, Street View, and Earth). For more information, see our copyright guidelines: http://journals.plos.org/plosone/s/licenses-and-copyright.

2.1. You may seek permission from the original copyright holder of Figures 1, 8, 9 to publish the content specifically under the CC BY 4.0 license.

2.2. If you are unable to obtain permission from the original copyright holder to publish these figures under the CC BY 4.0 license or if the copyright holder’s requirements are incompatible with the CC BY 4.0 license, please either i) remove the figure or ii) supply a replacement figure that complies with the CC BY 4.0 license. Please check copyright information on all replacement figures and update the figure caption with source information. If applicable, please specify in the figure caption text when a figure is similar but not identical to the original image and is therefore for illustrative purposes only.

Reviewers' comments:

Reviewer's Responses to Questions

**Comments to the Author**

1. Is the manuscript technically sound, and do the data support the conclusions?

Reviewer #1: Yes

Reviewer #2: Yes

2. Has the statistical analysis been performed appropriately and rigorously? 

Reviewer #1: Yes

Reviewer #2: Yes

3. Have the authors made all data underlying the findings in their manuscript fully available?

Reviewer #1: Yes

Reviewer #2: No

4. Is the manuscript presented in an intelligible fashion and written in standard English?

Reviewer #1: Yes

Reviewer #2: Yes

5. Review Comments to the Author

Reviewer #1: Review of P-one-D-20-0542

This study provides a detailed methodological approach to small-area-estimation of poverty based on data derived from mobile phones. Specifically, it addresses methodological advances in correcting how well the data derived from phones are able to assist in predictions of poverty rates. The study is well described and an important contribution to the field. I have several comments I hope can improve the general readability and accessibility of the manuscript, and one request for re-analysis that would improve applicability of the results.

As a general comment, I believe the authors have understated one of the main advances the paper offers, which is to allow for the idea that the statistical units at which poverty is measured may be best represented by more than one tower location. This idea should be better expressed in the introduction.

Both settlement weighting to derive augmented Voronoi polygons and radio-propogation-based modelling aim to address a core problem with how population density interacts with tower densities. While the results in the present study do not show significant improvements in predictive power,

I feel that it could be valuable to further explore the relative contribution of settlement weighting and radio-propagation modelling in the context of population density and tower distribution. For example, in the simulation studies, the predictive power of the model was higher in urban versus Rural areas. It would be of interest to see how this difference played out in Senegal.

In the conclusions, the author suggests that model misspecification could be the reason for a lack of significant model improvement using propogation models. A further potential interpretation – and possible contribtion of the study is that it hints at a lower limit to the scale at which spatial heterogeneity in poverty rates can be discerned using CDR data, at least in the context of poverty data aggregated within statistical units.

Please be consistent with capitalisation of ‘Voronoi’

Line 196: Please define HATA the first time you use it in text. Or does HATA refer to the author in citation 53?

Line 332: Please define BSA as Best server area here.

Line 350: please define IDW here as inverse distance weights

Results

Line 596: I don’t see any differentiation between urban and rural predictions in the Senagal case. Can these be provided for interest?

Figure 2: please be consistent in your labelling of the different scenarios so that 2e and 2f are labelled BSA and IDW in line with other text and tables.

Reviewer #2: Please see attached

Please see attached

Please see attached

Please see attached

Please see attached

Please see attached

Please see attached

Please see attached

Please see attached

6. PLOS authors have the option to publish the peer review history of their article (what does this mean?). If published, this will include your full peer review and any attached files.

Reviewer #1: No

Reviewer #2: No

---

## [Author Response · Author response to Decision Letter 0]

20 Aug 2020

I am grateful to the academic editor and the two reviewers for their constructive and excellent comments. These have been very helpful for improving and preparing the revised version of this paper. I have done my best to respond to all comments. The file 'Responses to Reviewers' show how I addressed each comment.

---

## [Decision Letter · Decision Letter 1]

26 Oct 2020

Better coverage, better outcomes? Mapping mobile network data to official statistics using satellite imagery and radio propagation modelling

PONE-D-20-05427R1

Dear Dr. Koebe,

We’re pleased to inform you that your manuscript has been judged scientifically suitable for publication and will be formally accepted for publication once it meets all outstanding technical requirements.

Kind regards,

Jacinto Estima

Academic Editor

PLOS ONE

Additional Editor Comments (optional):

Reviewers' comments:

Reviewer's Responses to Questions

**Comments to the Author**

1. If the authors have adequately addressed your comments raised in a previous round of review and you feel that this manuscript is now acceptable for publication, you may indicate that here to bypass the “Comments to the Author” section, enter your conflict of interest statement in the “Confidential to Editor” section, and submit your "Accept" recommendation.

Reviewer #1: All comments have been addressed

2. Is the manuscript technically sound, and do the data support the conclusions?

Reviewer #1: Yes

3. Has the statistical analysis been performed appropriately and rigorously? 

Reviewer #1: Yes

4. Have the authors made all data underlying the findings in their manuscript fully available?

Reviewer #1: Yes

5. Is the manuscript presented in an intelligible fashion and written in standard English?

Reviewer #1: Yes

6. Review Comments to the Author

Reviewer #1: (No Response)

7. PLOS authors have the option to publish the peer review history of their article (what does this mean?). If published, this will include your full peer review and any attached files.

Reviewer #1: No

---

## [Editor Report · Acceptance letter]

29 Oct 2020

PONE-D-20-05427R1 

Better coverage, better outcomes? Mapping mobile network data to official statistics using satellite imagery and radio propagation modelling 

Dear Dr. Koebe:

I'm pleased to inform you that your manuscript has been deemed suitable for publication in PLOS ONE. Congratulations! Your manuscript is now with our production department. 

Kind regards, 

on behalf of

Dr. Jacinto Estima 

Academic Editor

PLOS ONE